# Multiple Receptors Involved in Invasion and Neuropathogenicity of Canine Distemper Virus: A Review

**DOI:** 10.3390/v14071520

**Published:** 2022-07-12

**Authors:** Jianjun Zhao, Yanrong Ren

**Affiliations:** 1College of Animal Science and Veterinary Medicine, Heilongjiang Bayi Agricultural University, Daqing 163319, China; 2Heilongjiang Provincial Key Laboratory of Prevention and Control of Bovine Diseases, Heilongjiang Bayi Agricultural University, Daqing 163319, China; 3Institute of Special Animal and Plant Sciences, Chinese Academy of Agricultural Sciences (CAAS), Changchun 130112, China; renyanrong0927@163.com

**Keywords:** canine distemper virus, central nervous system, neuropathogenicity, neural receptor

## Abstract

The canine distemper virus (CDV) is a morbillivirus that infects a broad range of terrestrial carnivores, predominantly canines, and is associated with high mortality. Similar to another morbillivirus, measles virus, which infects humans and nonhuman primates, CDV transmission from an infected host to a naïve host depends on two cellular receptors, namely, the signaling lymphocyte activation molecule (SLAM or CD150) and the adherens junction protein nectin-4 (also known as PVRL4). CDV can also invade the central nervous system by anterograde spread through olfactory nerves or in infected lymphocytes through the circulation, thus causing chronic progressive or relapsing demyelination of the brain. However, the absence of the two receptors in the white matter, primary cultured astrocytes, and neurons in the brain was recently demonstrated. Furthermore, a SLAM/nectin-4-blind recombinant CDV exhibits full cell-to-cell transmission in primary astrocytes. This strongly suggests the existence of a third CDV receptor expressed in neural cells, possibly glial cells. In this review, we summarize the recent progress in the study of CDV receptors, highlighting the unidentified glial receptor and its contribution to pathogenicity in the host nervous system. The reviewed studies focus on CDV neuropathogenesis, and neural receptors may provide promising directions for the treatment of neurological diseases caused by CDV. We also present an overview of other neurotropic viruses to promote further research and identification of CDV neural receptors.

## 1. Introduction

The canine distemper virus (CDV) is a highly contagious pathogen that causes canine distemper and is characterized by fever, pneumonia, and most critically, leukopenia and neurologic deficit. This results in a severe and frequently lethal disease in a large number of carnivores [1]. In recent years, CDV cross-species infections were observed in nonhuman primates, including bred and wild rhesus monkeys and cynomolgus monkeys [2,3]. The CDV mortality rate in rhesus monkeys reaches 5–30% [3].

CDV belongs to the genus *Morbillivirus*, family *Paramyxoviridae*, which also includes the measles virus (MeV), peste des petits ruminants virus, rinderpest virus, phocine distemper virus, and cetacean morbillivirus [4,5]. The CDV genome is 15,690 nucleotides long and encodes six structural proteins: the nucleocapsid protein (N), phosphoprotein (P), matrix protein (M), fusion protein (F), hemagglutinin protein (H), and large protein (L) [6]. Two non-structural proteins, V and C, are encoded by the same gene as that encoding the P protein [7]. Both the V and C proteins are associated with the immunosuppression of the host. The V protein inhibits the nuclear translocation of the signal transducer and activator of transcription 1 (STAT1) and the signal transducer and activator of transcription 2 (STAT2), thus interfering with type I and type II interferon (IFN)-mediated transcriptional activation, with an antagonistic effect on innate immunity [8]. The C protein is necessary for CDV spread in the lymphatic organs. However, a C-defective virus may be ferried by lymphocytes through the lymphoid organs, with retention of its full immunosuppressive activity, which could result in host death [9]. The H and F proteins are virus envelope glycoproteins that play key roles in the recognition of virus cellular receptors and host cell entry. The H protein mediates viral binding to the cell membrane, and the F protein executes the fusion of viral and cell membranes, enabling the entry of the viral genome into the cytoplasm [10].

The signaling lymphocyte activation molecule (SLAM or CD150) expressed on activated T cells, immature thymocytes, memory T cells, B cells, activated monocytes, and dendritic cells [11], and the adherens junction protein nectin-4 (also known as PVRL4) expressed preferentially in the tracheobronchial epithelium are lymphatic and epithelial cell receptors for CDV recognition, respectively [12] (Figure 1). In the initial infection stages, the virus enters the respiratory tract and interacts with SLAM-positive immune cells, probably dendritic cells or alveolar macrophages. These cells then traffic to local draining lymph nodes, where the virus spreads to T and B cells [13]. The infected circulating immune cells subsequently disseminate the virus throughout the lymphatic system, which is followed by myeloid cell infiltration of the lamina propria and viral spread to epithelial tissues via contact with nectin-4, which is expressed on the basolateral side of the apical junction complexes in epithelial cells [14]. Finally, the virus invades the central nervous system (CNS), inducing severe neurological disease by establishing persistent infection [15]. Hence, SLAM recognition is essential for the CDV infection of the lymphatic tissue, viral dissemination, and two indicators of morbillivirus immunosuppression—the induction of leukopenia and the inhibition of lymphocyte proliferation activity [15]. By contrast, nectin-4 enables viral release from the apical side of the infected cell, enabling viral spread. Therefore, nectin-4-mediated epithelial cell infection is necessary for clinical disease and efficient viral shedding but not for immunosuppression. In addition, nectin-4 is associated with CDV neurovirulence [16,17].

Severe neurological symptoms usually occur in dogs with acute CDV infection, and approximately 30% of dogs show signs of neurological damage during CDV infection. CDV can establish a persistent infection in the brain, leading to old dog encephalitis and demyelinating leukoencephalitis [18]. CDV neuroinvasion occurs predominantly via the hematogenous route. CDV has been postulated to enter the dog brain hematogenously via infected lymphocytes that penetrate the blood–brain barrier (BBB) and subsequently release the virus into the cerebrospinal fluid (CSF), which results in periventricular and subpial lesions [18]. Once the virus invades the nervous system, neurological symptoms predominantly manifest as myoclonus, nystagmus, ataxia, postural reaction deficit, and tetraparesis or plegia [19]. CNS involvement is a complication that often occurs in parallel or subsequent to the infection of another organ [20]. The symptoms in the late phase are mainly mild conjunctivitis and CNS disturbances [21]. The clinical symptoms and neuropathy vary according to the neurological symptoms and pathological changes in the acute and late stages of CDV infection (Table 1). As shown by studies in ferrets, raccoon dogs, and dogs, CDV may invade via the anterograde pathway, utilizing the olfactory bulb as the primary viral target organ [22,23,24].

To date, SLAM and nectin-4 have not been detected in astrocytes; however, CDV infection of the CNS usually damages astrocytes [25,26]. Astrocytes play a key role in maintaining normal CNS physiology and critically control the response to brain injury and neurological diseases [18]. This indicates the presence of other, unknown receptors associated with CDV infection of the CNS. Indeed, an unknown CDV receptor on glial cells, named GliaR, has been confirmed [27]. This unknown glial cell receptor allows the noncytolytic cell-to-cell transmission of CDV among astrocytes [28]. The spread of CDV between glial cells may be dependent on the trans-synaptic mode, but its specific mechanism remains to be studied [28].

Measles, which is characterized by fever, conjunctivitis, and a maculopapular rash, is caused by MeV, a highly contagious human pathogen that rarely establishes persistent infection in the CNS. Only a few cases of measles develop subacute sclerosing panencephalitis (SSPE) after acute infection caused by the wild-type MeV genotypes [29]. Like CDV, MeV is a member of the genus *Morbillivirus* of the family *Paramyxoviridae*. The MeV H and F proteins are envelope glycoproteins that mediate receptor binding and membrane fusion, respectively. Similar to CDV receptors, the receptors for MeV are SLAM, expressed on immune cells [30], and nectin-4, expressed on epithelial cells [31]. The Edmonston MeV vaccine strain genotype A harbors many amino acid substitutions in the H protein, which allow it to exploit CD46 as an additional receptor [32]. Because SLAM and nectin-4 are not expressed on human neurons, it is likely that wild-type MeV uses other receptors to enter the CNS and cause SSPE.

In this review, we summarize (1) the recent research findings on CDV receptors and their role in canine distemper pathogenesis; (2) CDV invasion of the CNS and the underlying pathogenic mechanism; and (3) the possible neural receptors of CDV. This review of CDV infection of the CNS and the associated neural receptors highlights targets for the treatment of CDV-related neurological symptoms and the optimization of CDV vaccines.

## 2. Cell-to-Cell Fusion Mediated by CDV H/F Protein Complex

CDV binds to its receptors through the H and F proteins and invades the host cell by membrane fusion [33]. Membrane fusion begins when the H protein tetramer forms a complex with F trimers, which then binds to SLAM/nectin-4 [29]. Receptor binding induces conformational changes in the H tetramer. This creates a fusion-competent microenvironment [29]. The F trimers are then released and undergo an irreversible structural rearrangement, transforming into a postfusion state and forming fusion holes on the cell membrane surface, thus allowing viral entry [34]. The molecular integrity of the tetramer itself remains unchanged during receptor binding and F triggering. After the formation of the F protein trimer, a central “pocket” in the globular head domain of the F protein regulates the stability of its metastable prefusion conformational state [35]. The extent and efficiency of membrane fusion depend on the binding of the H protein to SLAM. That is because residues critical for membrane fusion are located in the lateral region of the H head (blades 4–6 of the beta-propeller) and align with the front site of the variable (V) domain of SLAM and because of the key role of residue E123 of SLAM, which is located within the front site of SLAM and may transmit a productive fusion-triggering signal to the H head domains [36]. A recent study demonstrated a substantially impaired fusion promotion by most H-stalk variants harboring alanine substitutions in the 126–138 “spacer” section [34]. Probing with an anti-CDV-H monoclonal antibody, which targets the linear H-stalk segment (residues 126–133), revealed that the spacer can effectively inhibit membrane fusion without interfering with H-receptor binding or F-interaction [34]. This indicates that the amino acids in the spacer may determine the efficiency of membrane fusion. The following conditions are necessary to complete membrane fusion: (1) proximity of the virus particle and the cell membrane, afforded by a continued interaction of the H protein with the receptor; (2) the aggregation and local assembly of multiple activated H/F complexes interacting with receptors; and (3) the presence of an additional physical force originating from the H/receptor-driven local curvature in the opposing donor and target membranes [37].

## 3. Receptors That Enable CDV Entry and Spread

### 3.1. SLAM, a Lymphocyte Receptor Involved in Host Immune Suppression

SLAM is a member of the immunoglobulin (Ig) superfamily that, in humans and mice, is expressed on thymocytes, activated lymphocytes, mature dendritic cells, macrophages, and platelets [38]. The extracellular region of SLAM contains a variable (V) domain and a constant (C2) Ig-like domain. The V domain is required for recognition and binding to the CDV H protein [36,39]. SLAM participates in various immune functions, including the costimulation of T cells and B cells, the secretion of IFN by Th1 cells, and the inhibition of B-cell apoptosis [40]. Using SLAM as a receptor, CDV primarily replicates in lymphocytes and macrophages in the respiratory tract and then propagates in the lymph nodes before disseminating in the body [15]. The binding of SLAM to CDV H proteins is necessary for the establishment of systemic infection. In one study, the blocking of this binding reduced the number of infected animals with viremia; by contrast, the effect was not apparent when the binding of the H protein to nectin-4 was blocked [15]. This indicates that the SLAM and CDV H protein binding is associated with immune suppression of the host. A compensatory mutation analysis confirmed that CDV isolates are subject to the selective pressure of SLAM-dependent cell entry. This selective pressure also operates during natural infection. A strong in vivo selective pressure drives the *H* gene towards the efficient interaction of the H protein with SLAM, selecting for compensating mutations in or near the SLAM-interacting surface on the H protein. Of note, suboptimal SLAM interactions lead to inefficient lymphocyte infection, which interrupts the transmission cycle [15]. Although SLAM is the principal receptor for CDV entry, SLAM expression in the brain has not been observed, and positive staining for SLAM is only detected in the cells of the vascular wall [28]. This suggests the presence of other receptors allowing CDV entry into the nervous system.

### 3.2. Nectin-4, an Epithelial Cell Receptor Associated with CDV Neurovirulence

Nectin-4, recently identified as an epithelial receptor for members of the *Morbillivirus* genus, is a member of the nectin family of adhesion molecules, which belong to the Ig superfamily [41,42,43]. Some members of the nectin family also function as entry receptors for other viruses. For example, nectin-like molecule 5 (CD155) is a receptor for poliovirus [44]. Nectin-4 contains an ectodomain with three Ig-like domains (V, C, and C), a transmembrane region, and a cytoplasmic tail [45]. Nectin-4 is expressed in bronchial, bronchiolar, gastric, and intestinal glandular epithelial cells as well as transitional epithelial cells, renal pelvis epithelium, tonsil epithelium, epidermis keratinocytes [31], and a variety of nerve cells (neurons, ependymal cells, etc.) in the CNS [27]. It is also used as a tumor cell marker that is highly expressed in embryonic cells, such as placental cells, and is expressed at low levels in the oral mucosa, nasopharynx, and lung [46].

Nectin-4 has been biochemically shown to bind to the CDV H protein via its V domain, leading to viral entry [47]. Infected immune cells are assumed to transmit CDV to airway epithelial cells via nectin-4 receptors located on their basolateral surface. Ultimately, these infected cells release the virus from the apical cell surface, enabling further infection and disease spread [15,48]. CDV is also transmitted to the CNS via the hematogenous pathway, i.e., in virus-infected lymphocytes that cross the BBB [49] and the CSF barrier [50]. Nectin-4 is not expressed at detectable levels in primary astrocytes and is undetectable in the white matter in the canine brain [26]. However, Pratakpiriya et al. [27] evaluated the distribution of CDV antigens in canine brain tissues and suggested that nectin-4 contributes to CDV neurovirulence. In addition, using the CDV-induced demyelinating leukoencephalitis (CDV-DL) model, Lempp et al. [20] reported that oligodendrocytes are infected by CDV to a lesser degree than astrocytes and that the infection may be associated with demyelination. The authors also observed a large number of infected astrocytes not expressing SLAM and nectin-4 on their surface, indicating the existence of other receptors that medicate CDV entry into the nervous system.

### 3.3. GliaR, an Unknown Glial Cell Receptor

In addition to SLAM and nectin-4, other molecules were found in association with CDV infection. Both the CDV F and H proteins bind to immobilized heparin sulfate molecules; however, one study demonstrated that the infection of B95a cells by CDV is mainly mediated via the high-affinity SLAM receptor, whereas the contribution of heparin-like molecules appears low [51]. However, the virus can infect cells not expressing the high-affinity receptor via another pathway in which heparin-like molecules are involved [51]. CD9, a tetraspan transmembrane protein, is involved in regulating virus-induced cell fusion [52]. It acts as an inhibitor of F-protein-mediated membrane fusion or the H–F or H–F–M protein interactions [53,54]. Notably, a recent study revealed a 57 kDa molecule among the membrane proteins of chicken embryo fibroblasts that was different from the receptors present on lymphocytes and HEK-293 cells, indicating the possibility that it acts as a receptor involved in the CDV infection of chicken embryo fibroblasts [55]. Further, CD46, a receptor for laboratory-adapted and attenuated MeV that allows virus–cell binding and fusion, could be an alternative receptor for MeV in SLAM-negative canine cells [56,57]. While CD46 has been detected in canine neoplastic lymphoid cells, it does not act as a receptor of wild-type or laboratory-adapted CDV [40,58].

Recently, the presence of a heretofore unknown receptor, not nectin-4, expressed on astrocytes was confirmed in association with the neurovirulence of CDV [27]. Despite nectin-4 not being expressed in glial cells, CDV infection of the CNS usually damages these cells. Therefore, the possibility of the presence of other receptors on the surface of glial cells should be explored. Of note, the infection of cultured primary dog brain cells with a bioengineered “nectin-4–blind” recombinant CDV strain that exhibits membrane fusion activity is limited to the formation of microfusion pores, which may depend on specific conditions, including the participation of receptors other than nectin-4 [26]. Further, while the CDV-DL strain triggers limited cell fusion of Vero cells, the nectin-4-blind recombinant CDV strain retains complete cell-to-cell transmission function in astrocytes. Hence, the persistence of CNS infection depends on noncytolytic cell-to-cell spread, which does not depend on SLAM and nectin-4 expression in astrocytes, suggesting the existence of a third receptor expressed in glial cells (provisionally referred to as GliaR). GliaR controls the cell-to-cell transfer of the CDV nucleocapsid, allowing the virus to continuously infect the brain and cause complications [26].

Accordingly, we propose a model that explains the sustained CDV neuroinvasion and the ensuing immunopathological complications. According to the model, CDV begins to replicate in lymphoid tissues by binding to SLAM receptors. The infection then spreads to epithelial tissues when the virus binds to nectin-4. The putative GliaR receptor expressed on perivascular astrocytes is accessed either by infected endothelial cells or CDV-carrying circulating mononuclear cells, which can penetrate the BBB [26]. Binding to the nectin-4 receptor also enables the infected endothelial cells and monocytes to break through the ependymal and meningeal barriers of the CNS, allowing contact between CDV and the underlying astroglial cell layer [26]. Once astrocytes are targeted, the infection spreads further into the white matter in a GliaR-dependent manner, subsequently leading to neurological complications. While SLAM and nectin-4 are essential for understanding the pathogenesis of systemic CDV infection, the presence of a third receptor on brain cells—which mediates the transfer of viral nucleocapsids between cells—would explain the neurological complications caused by CDV infection.

Morbilliviruses, particularly MeV and CDV, cause neurological diseases, but the neuronal receptor has not yet been identified. MeV spreads from epithelial cells to primary neurons via nectin-elicited cytoplasm transfer (NECT) [59]. In NECT, in addition to transferring transmembrane proteins, the virus takes advantage of cytoplasm flow to spread [59]. Intercellular protein transfer relies on cell–cell contacts established by the nectin-adhesive interface and the nectin cytoplasmic region. The neuronal uptake of virus particles can occur in any innervated epithelial tissue that expresses nectin-4, such as the nasal turbinate [60].

The infection of neuronal cell bodies requires the transfer of infectious particles into the isolated axons, followed by transport to the cell body. Subsequently, virus particles reach the brain by retrograde transport. The virus particles remain functional after uptake to drive the spread of infection. In one study, the H protein and an F protein variant harboring fusion-enhancing substitutions were shown to be crucial for membrane fusion and subsequent MeV transmission between neurons [61]. Specifically, the T461I substitution in the F protein conferred enhanced fusion activity and contributed to MeV spread in neurons. This hyperfusogenic virus spread in cells lacking SLAM and nectin-4 [62] and infected the CNS, causing lethal disease [63]. The above studies open up new avenues for researching as yet unidentified CDV receptors and the underlying neuropathogenesis mechanism(s).

## 4. CDV Invasion and Pathogenicity in the CNS

Based on studies with a recently developed mouse-adapted neuroinvasive CDV Onderstepoort strain that causes neuropathogenicity in mouse, the virus first destroys the BBB and then reaches the CNS, causing various neurological symptoms [49]. CDV primarily infects neuroependymal cells lining the ventricular wall and neurons of the hippocampus and cortex adjacent to the ventricle. Subsequently, the virus extensively infects the brain surface and then the parenchyma and the cortex. CDV spreads in a unidirectional retrograde manner along the neuronal processes in the hippocampal formation, i.e., from the CA1 region to the CA3 region and the dentate gyrus [49]. CDV infects a family of growth-promoting glial cells, including specialized macroglia with Schwann-cell-like structures, promoting the infection of the nervous system [49]. Further, CDV with the wild-type genotype enters the brain through the olfactory system in ferrets, with transneuronal transmission along the olfactory axons [22]. In the nasal cavity, most CDV-positive cells reside in the submucosa and include fibroblasts, large macrophage-like cells, and lymphocyte-like cells [64]. The olfactory bulb is the first location of macroscopic expression of the virus antigen. The neurons located in the olfactory mucosa, along the olfactory nerve filaments passing through the cribriform plate and into the olfactory glomeruli, thereby constitute the synapse between the olfactory nerve fibers and mitral cells. The virus is transmitted across neuronal synapses and spreads anterogradely to deeper CNS structures [65]. At the same time that the CDV infects most olfactory nerve fibers and spreads to the olfactory glomerulus, the number of virus-antigen-expressing cells associated with the choroid plexus and blood vessels also increases. This suggests a direct hematogenous dissemination route [50] wherein the viral particles enter the circulating CSF and spread to the lining of the ventricles, pia mater, and underlying molecular layers of the cerebral and cerebellar cortex; invade the brain parenchyma surrounding the blood vessels; migrate from the olfactory glomeruli to mitral cells and further towards the olfactory cortex; and then infect neurons and glial cells, allowing the virus to spread to deeper brain tissues (Figure 2) [66].

Based on experiments with CDV-infected ferrets, the disease duration associated with different CDV isolates is the main neurovirulence factor that determines the extent of CNS infection. The highly virulent CDV 5804P causes a rapidly progressing disease (less than 2 weeks from infection to death) characterized by a complete loss of immune system function and death, with no neurological symptoms during the disease process [67]. In contrast, the disease caused by CDV A75/17 usually persists for between 3 and 5 weeks, and most animals develop neurological signs. Despite a widespread infection of immune and epithelial tissues, residual immune function is maintained in these animals [68]. By exchanging the H proteins of CDV 5804P and A75/17 and assessing the pathogenesis of the chimeric viruses in ferret, the authors observed that both H proteins support neuroinvasion and the subsequent development of clinical neurological signs if given enough time, indicating that disease duration determines the extent of CNS infection [67]. The lack of neurological signs in CDV-5804P-infected ferrets might be associated with the rapid disease progression in other organs rather than the inability of the virus to infect CNS cells. These observations also demonstrate that the H protein from a neurovirulent isolate (A75/17) is more efficient at mediating CDV neuron invasion than the H protein from 5804P, probably because of a high affinity for the yet-to-be-identified receptor on neural cells. In addition, the dynamic interplay between the virus and the host immune system, which leads to differences in the duration of the disease caused by different CDV strains, may also explain the spectrum of CNS diseases associated with CDV [67].

CDV antigens have been detected in a variety of cell types in the CNS, including neurons, astrocytes, ependymal cells, and olfactory ensheathing cells (OECs), among others, and were shown to induce eosinophilic inclusion body formation and neuronophagia (Figure 3) [23]. The neurological complications observed in MeV and CDV infections result from viral persistence arising from viral cell-to-cell spread through neurons in SSPE and through astrocytes in demyelinating distemper encephalitis [28,61]. CDV antigens and RNA are detected in neurons in all types of nervous distemper during the early phase and are most prominent in distemper polioencephalitis [21]. Despite viral spread, there is little evidence of MeV-induced cell death, syncytium formation, or infectious virus production in neurons, while MeV RNA continues to persist in the CNS [61,69]. Since neurons, which are important cell targets affected in SSPE, express neither SLAM nor nectin-4, CDV is thought to exploit a different infection mechanism that does not involve these two receptors [62]. Early research on the spread of MeV between neurons indicated that MeV adopts a trans-synaptic mode of spread and does not require CD46 expression [61,70]. However, viral transport across the synaptic cleft requires membrane fusion, with neurokinin-1 (NK-1)—a highly conserved protein expressed in diverse mammalian cells [71]—playing a possible key role as an MeV-F receptor. Therefore, we speculate that CDV may also infect and spread in neurons using the trans-synaptic mode with the help of an unknown receptor.

Unlike astrocyte infection, oligodendroglial infection is relatively rare during demyelination. Although oligodendrocytes can be directly infected with CDV, only a few cells carry the virus particles [21]. While CDV transcription in oligodendrocytes is not defective, viral proteins are not synthesized in these cells; therefore, CDV infection of oligodendrocytes is restricted and characterized by the presence of CDV nucleic acid and the lack of viral antigens [21]. CDV-DL infection does not necessarily lead to oligodendroglial death but may induce oligodendrocytic dystrophy. Oligodendrocyte degeneration results from CDV-induced microglial cell activation [72]. Some infected oligodendrocytes are hypertrophic, microvacuolated, and show organelle loss, leading to oligodendrocyte metabolic disorders. This is followed by decreased transcription of the myelin gene, which subsequently results in demyelination; however, the underlying mechanism is not yet fully understood.

CDV strain-dependent differences in brain tissue damage and lesions in the infected host have been investigated. The Snyder Hill strain primarily causes acute polioencephalitis, whereas CDV A75/17 and R252 predominantly cause demyelinating leukoencephalitis [24]. The CNS lesions caused by the Snyder Hill strain are multifocal and form in both the grey matter and white matter. The lesions in the grey matter are severe and extensive, whereas inflammation in the white matter is light [24]. This can be followed by gliosis and neuronal degeneration. However, CDV A75/17 predominantly causes demyelinating leukoencephalitis, with the lesions being most prominent in the periventricular and subependymal white matter areas of the midbrain and cerebellum [24]. Neuronal injury and neuronophagia are rare, except for focal granule cell necrosis in the cerebellar cortex [24]. Demyelination appears as vesiculation and a progressive pallor of the white matter [24]. Further, CDV A75/17 causes acute disease with neurological signs in 10% of infected animals. However, neurovirulent CDV R252 generally results in nonsuppurative encephalomyelitis with mild or subclinical disease and persistent encephalitis [24]. The CNS lesions induced by CDV R252 are centered on myelinated tracts, mainly at the level of the fourth ventricle [24]. While the Snyder Hill strain readily infects neurons, CDV R252 and A75/17 infect neurons poorly and show a stronger tropism for astroglia, with the disease caused by CDV A75/17 and R252 persisting much longer than that caused by the Snyder Hill strain [24].

Different CDV strains can infect OECs and Schwann cells, albeit to different degrees [73]. The consequences of infection mainly depend on the virulence of the respective CDV strain, the age of the infected individual, and their immune status [20]. A rapid and fatal course of disease is correlated with persistent viremia and a lack of virus-neutralizing antibody in the serum [74]. A failing or insufficient humoral response during the infection period may promote secondary viremia, while the presence of a robust antiviral immune response may enable the infected individual to eliminate the virus, resulting in recovery [74]. Immature dogs are more prone to developing acute disease than mature dogs, while mature dogs usually develop chronic encephalomyelitis [19]. While chronic lesions in the CNS predominate in immunocompetent individuals, acute CDV encephalomyelitis is relatively common in severely immunosuppressed individuals.

SLAM receptors often exhibit species differences, resulting in differences in CDV tropism. Nectin-4 is also a common CDV receptor, but critical residues of the H protein required for nectin-4 binding are highly conserved [39]. Considering the potential differences in CDV infectivity and pathogenicity based on viral–host cell interactions, further investigation into host cell receptors in a variety of CDV-susceptible carnivores is warranted [39]. Further, CDV can adapt to related host species via mutations in the receptor-binding region of the H protein for enhanced virulence. Single amino acid substitutions in the H protein, such as D540G or Y549H, have been linked to enhanced fusion activity in cells expressing human SLAM and viral adaptation to new hosts [75]. The wild-type A75/17 CDV establishes a persistent infection in the CNS but infects the cells very inefficiently, with a very limited cytopathic effect, and the virus spreads in a cell-to-cell manner without obvious syncytium formation. In contrast to the wild-type strains, the Onderstepoort vaccine strain, which has been extensively isolated in vitro, produces a pronounced CPE in many cell types, accompanied by the formation of large syncytia. Further, while the H protein of the neurovirulent CDV A75/17 drives persistent infection in a SLAM-dependent manner, the F protein reduces the cell-to-cell fusion independent of SLAM, demonstrating that the F protein plays a key role in determining persistent viral infection [76].

## 5. Anti-CDV Immune Response in the CNS

In the acute phase of CDV infection, there is a complete lack of an effective antiviral neutralizing immune response. Acute noninflammatory lesions are characterized by demyelination with spongy vacuolation of the white matter, reactive gliosis, the progressive infection of astrocytes, the swelling of endothelial nuclei, vascular proliferation, and an increased number of microglial cells and macrophages [77]. Chronic active lesions exhibit various degrees of inflammation, such as mononuclear perivascular cuffing, extensive invasion of the parenchyma by inflammatory cells (including macrophages), severe damage of the white matter, and necrosis, or even hemorrhage, of several demyelinated areas [20]. CNS infections are often accompanied by demyelination, which usually leads to lesions in the cerebral cortex, hippocampus, thalamus, brainstem, and—to a lesser extent—the cerebellar cortex [78].

The persistence of MeV and CDV leads to the neurological complications that result from viral cell-to-cell spread through neurons in SSPE and through astrocytes in demyelinating distemper encephalitis. Demyelination, a typical pathological change caused by CDV infection in the brain, coincides with the CDV infection of astrocytes [28]. Compared with uninfected astrocytes, astrocytes with CDV-DL have increased metabolic activity, an increased number of mitochondria, an increased rough endoplasmic reticulum size, and a highly active Golgi apparatus with many associated vesicles. One peculiar feature of CDV-DL is the viral colonization of vimentin-positive astrocytes, which are a population of immature and/or reactive astroglial cells that may support CDV persistence and spread within the brain of chronically infected dogs [20]. CDV-DL is characterized by lesions with variable degrees of demyelination and mononuclear inflammation, accompanied by the dysregulated orchestration of cytokines.

The activation of different cells in the CNS and cytokine secretion constitutes the antiviral immune response of the CNS [79]. In the case of MeV infection, CD8^+^ T lymphocytes are among the first mononuclear cells to arrive at the infection site, after which an increasing number of inflammatory cells, including macrophages/microglia and antibody-secreting plasma cells, are recruited [80]. CD4^+^ T lymphocytes overcome the immunosuppressive milieu of the CNS by secreting pro-inflammatory tumor necrosis factor (TNF)-α and IFN-γ [81]. Indeed, a semiquantitative RT-PCR analysis revealed a prominent upregulation of the pro-inflammatory cytokines interleukins 6, 8, and 12 and TNF-α in early distemper CNS lesions [82]. Furthermore, the local production of IFN-α/β induces the expression of major histocompatibility complex [83] antigens in CNS cells. After phagocytosis, microglial cells become more active, upregulate major histocompatibility complex molecules, acquire antigen-presentation capability, and secrete chemokines [84]. This initiates the upregulation of adhesion molecules on adjacent endothelial cells of the BBB. TNF-α is predominantly expressed by astrocytes and may play a crucial role in the pathogenesis of early demyelination. IFNs contribute to the antiviral immune response; however, in astrocytes, the presence of IFN-α/β receptor (IFNAR) signaling exacerbates astrogliosis, and the activation of astrocytes is reinforced via the IFN-I response. Therefore, the IFN-I response in astrogliosis may control CDV propagation throughout the CNS. In addition, the secretion of pro-inflammatory cytokines is accompanied by the polarization of microglia/macrophages towards the neurotoxic, classically activated M1 phenotype [85]. An analysis of cytokine production and expression following infection with MeV revealed that some proteins are upregulated without IFNAR signaling during hippocampal astrocyte proliferation [86]. The role of other extracellular matrix proteins in the pathogenesis of demyelinating leukoencephalitis has also been confirmed [86].

## 6. Neurotropic Viruses in Humans and Animals and Their Receptors

In addition to CDV and MeV, other neurotropic viruses that cause acute infections in humans and animals, including the Japanese encephalitis virus (JEV) [87,88], West Nile virus (WNV), dengue virus, yellow fever virus, tick-borne encephalitis virus [89], influenza virus, and rabies virus (RABV), have been identified. The viruses responsible for latent infections include the herpes simplex virus (HSV) and varicella-zoster virus, while those causing slow virus infections include the JC polyomaviruses (JCV) and retroviruses, such as human T-lymphotropic virus 1 (HTLV-1) [90] and the human immunodeficiency virus [91,92]. Most influenza virus strains are considered non-neurotropic, but some neurotropic influenza strains (such as H1N1) can induce cumulative neuroinflammatory damage, leaving the brain less resilient to future insult and contributing to neurodegeneration [93]. These viruses enter the nervous system through different mechanisms and cause different types of neuropathies.

The invasion of the CNS by neurotropic viruses is mainly accomplished via two pathways: the hematogenous dissemination route and [44] the anterograde pathway through the olfactory nerve (Figure 3). The hematogenous pathway mainly involves the destruction of the BBB to allow CNS entry. During JEV infection, JEV-infected endothelial cells upregulate the expression of ICAM-1 and cytokine-induced neutrophil chemoattractant 1 (CINC-1), which participate in leukocyte trafficking into the CNS [94]. This mechanism recruits immune cells into the brain, leading to BBB disruption. In RABV infection, pathogenesis begins after viral entry into the skin or the mucous membranes, where the virus begins replication in the myocytes, invades the local sensory and motor neurons at the site of the bite, and migrates towards the CNS through sensory and motor axons via a fast axonal retrograde transport system [95]. The permeability of the BBB is enhanced by the reduced expression of tight junction [96] proteins, which promotes inflammatory cell penetration of the CNS. HIV and mouse adenovirus type 1 infections may also disrupt the BBB by influencing the expression of tight junction proteins. A different mechanism of BBB disruption has been described for the Nipah virus (NiV), which is similar to that of CDV. Specifically, NiV-induced endothelial damage is observed in peripheral blood mononuclear endothelial cells, leading to a compromised BBB and penetration of the small brain vessels by leukocytes. The pathogenesis of NiV infection is mostly attributed to endothelial destruction, multinucleated syncytia, vasculitis-induced thrombosis, ischemia, and microinfarction in the CNS. Subsequently, the neurons and glial cells in the brain parenchyma are infected, allowing the virus to overcome the BBB [87]. Overall, viruses such as JEV, WNV, HSV, and HIV-1 disrupt the BBB using a “Trojan horse” mechanism where the virus is carried into the brain by infected inflammatory cells via the overexpression of adhesion molecules or by altering tight junction proteins to allow the penetration of the CNS by inflammatory cells.

Some neurotropic viruses, including HSV-1, the vesicular stomatitis virus, the Borna disease virus, RABV, the influenza A virus, the parainfluenza virus, and prions, can enter the CNS through the olfactory route [96,97]. Similar to CDV, HSV first infects the olfactory receptor neurons in the olfactory mucosa, after which the virus is anterogradely transported to the olfactory bulb. In addition, picornaviruses, WNV, and NiV can enter the CNS via the channels formed by OECs, which form an open connection with the CNS [98]. As for RABV and HSV, virus particles are released into the synaptic cleft and internalized [99]. In the case of MeV, the virus does not bud from infected neurons or form syncytia; however, the presence of nucleocapsids in the axons and at the presynaptic membranes of infected neurons suggests a contact-dependent, trans-synaptic spread of MeV [100]. The spread of CDV in most cells also relies on this process, although CDV is spread in a noncytolytic manner among astrocytes. Another possible mechanism is the non-trans-synaptic cell-to-cell transmission of viruses in solid tissues, in which the viruses released from infected cells infect neighboring cells. RABV and HSV particles have been observed not only in the synaptic cleft but also close to the cell body, confirming this mechanism [99].

After JEV infection, macrophages are the main infected cells in the brain tissue. The lectin receptors expressed on macrophages interact with the virus and play an important role in JEV-induced lethality. JEV then interacts with the C-type lectin superfamily member 5 (CLEC5A) expressed on the cell membrane and induces DAP12 phosphorylation in macrophages. Upon CLEC5A activation, pro-inflammatory cytokines are secreted, and the antiviral response is finally activated [87]. Receptors for RABV glycoprotein (G) include the neural cell adhesion molecule (NCAM), p75 nerve growth factor receptor (p75NTR), and nicotinic acetylcholine receptor (nAchR), which are responsible for the neurotropism of RABV. p75NTR, located in the endoneurium of Schwann cells, facilitates RABV infection of the CNS, whereas nAchR, located on the postsynaptic membrane of the neuromuscular junction, can enrich RABV particles at the neuromuscular junction to promote the infection of motor neurons and sensory neurons [101]. Once RABV binds to these receptors via its G protein, a neutralizing antibody response is induced. Following internalization, the G protein mediates the fusion of the viral envelope with the endosomal membrane. Then, after crossing the BBB, the virus replicates in neurons and glial cells to finally infect the CNS [87].

HIV invasion of the nervous system usually leads to acquired immune deficiency syndrome (AIDS) encephalitis. HIV enters human embryonic microglia through a receptor on CD4^+^ T lymphocytes, and the infection can further spread in the CNS via the contact of glial cells with infected CD4^+^ T lymphocytes [102,103]. However, the massive replication of the virus requires synergy among chemokines [104].

Neurological complications in individuals infected with severe acute respiratory syndrome coronavirus 2 (SARS-CoV-2) have not been widely reported. Nonetheless, cases where the SARS-CoV-2 infects the nervous system and causes encephalitis have been reported [105,106]. Neurons have been recently demonstrated to be a target of SARS-CoV-2, with devastating consequences, such as localized ischemia in the brain and cell death. Single-cell RNA-seq confirmed that SARS-CoV-2 infection induces a locally hypoxic environment in neuronal regions. Furthermore, angiotensin-converting enzyme 2 (ACE2) is required for SARS-CoV-2 infection of brain organoids, as evidenced in anti-ACE2 blocking assays [107]. Recent studies also showed that, in addition to ACE2, other receptors may be associated with SARS-CoV-2 infection of the nervous system. For instance, a screening analysis revealed that the knockdown of low-density lipoprotein receptor class A domain containing 3 (LDLRAD3) dramatically reduces SARS-CoV-2 infection in neuron cells, suggesting its critical function in mediating viral entry into neurons [108]. LDLRAD3 is a member of the low-density lipoprotein scavenger receptor family that is highly expressed in neurons and has been reported to regulate amyloid precursor protein in neurons [109]. Nevertheless, the SARS-CoV-2 mechanism of infection of the nervous system remains to be explored further.

## 7. Future Perspectives

Viral persistence in canine distemper is favored by two factors: (1) a noncytolytic strategy of virus replication and spread and the limited release of virus material into the extracellular space, which may delay viral detection by the local immune response, and (2) the limited translation of viral RNA in certain cells, such as oligodendrocytes and neurons [44,110]. These two factors may be related to the configuration and processing of the F protein, although this requires further confirmation. Studies of the MeV infection of neurons showed that the infection process relies on the participation of the H and F proteins [111,112]. MeV isolated from SSPE patients harbors a fusion-enhancing T461I substitution in the F protein that contributes to MeV spread in human neurons and NT2N cells lacking SLAM and nectin-4. The H protein is also required for the spread of the hyperfusogenic MeV between neurons, indicating the presence of other neuronal receptors concentrated at synapses that interact with the MeV H protein [61,62]. We speculate that this hyperfusogenic F-protein-dependent mechanism of invasion and spread in the CNS via neuronal receptors may also be applicable to CDV. Hence, blocking membrane fusion affected by hyperfusogenic F protein might be a putative therapy for fatal CDV infection in the CNS.

CDV can infect a variety of nerve cells in the CNS via as yet unidentified neural receptors, such as GliaR, which is related to glial cell infection. However, issues remain that require further exploration, such as the receptor on the astrocyte surface that has not yet been fully characterized. Further, host molecules, cytokines related to the nervous system infectivity of CDV, and their functions have not yet been fully explored. The identification of the host gene targets required for viral infection may provide promising candidates for pharmacological inhibition. Future studies focusing on CDV neuropathogenesis may provide promising directions for the treatment of the neurological diseases caused by CDV.

## Figures and Tables

**Figure 1 viruses-14-01520-f001:**
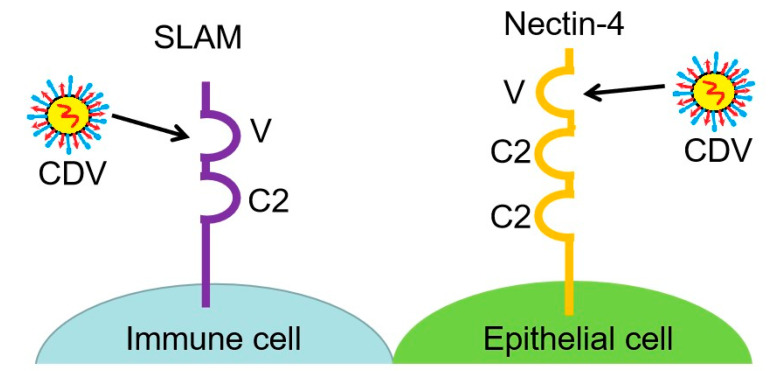
Interaction between canine distemper virus (CDV) and its immune and epithelial cell receptors, SLAM and nectin-4, respectively. CDV binds to the variable (V) domain of SLAM or nectin-4 on immune cells or epithelial cells, respectively.

**Figure 2 viruses-14-01520-f002:**
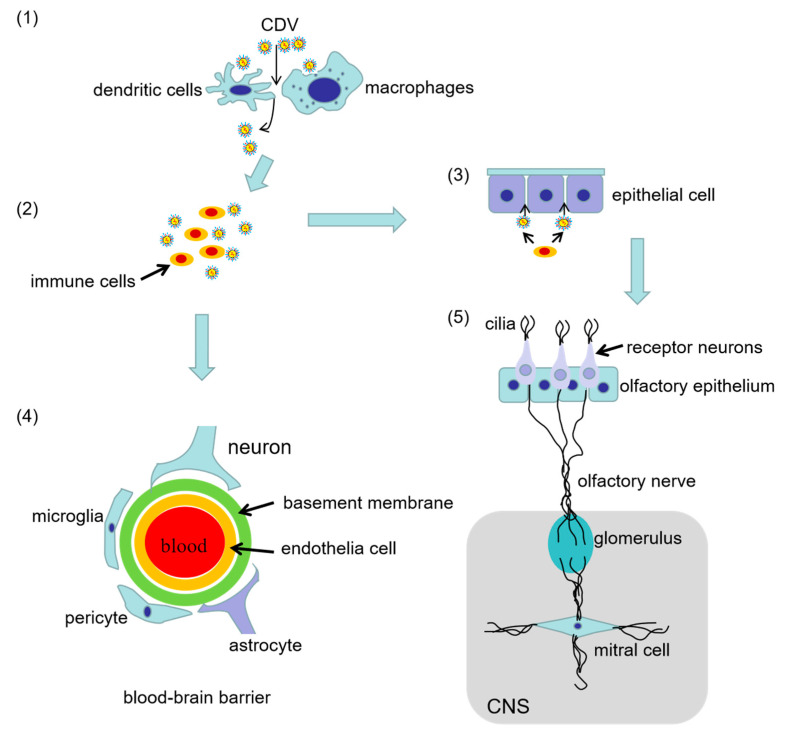
Diagram illustrating two CDV entry pathways into the CNS. Initially, CDV invades and spreads in airway immune cells (macrophages or dendritic cells) by binding to SLAM (**1**). CDV infects immune cells in the lymph nodes by binding to SLAM and leads to primary viremia and immunosuppression (**2**). Later, the virus spreads to epithelial cells expressing nectin-4 via infected immune cells (**3**). In the proposed hematogenous dissemination pathway, CDV induces endothelial damage, leading to a compromised BBB and the penetration of brain vessels by leukocytes. Then, the virus exits the periphery and infects the central nervous system (CNS) by infiltrating the BBB. The BBB is composed of brain microvascular endothelium cells with specialized tight junctions surrounding the basement membrane, pericytes, astrocytes, and neurons (**4**). The anterograde pathway through the olfactory nerve involves invasion via the olfactory epithelium and olfactory neurons. CDV spreads to the CNS by anterograde axonal transport along the olfactory nerve into the brain (**5**).

**Figure 3 viruses-14-01520-f003:**
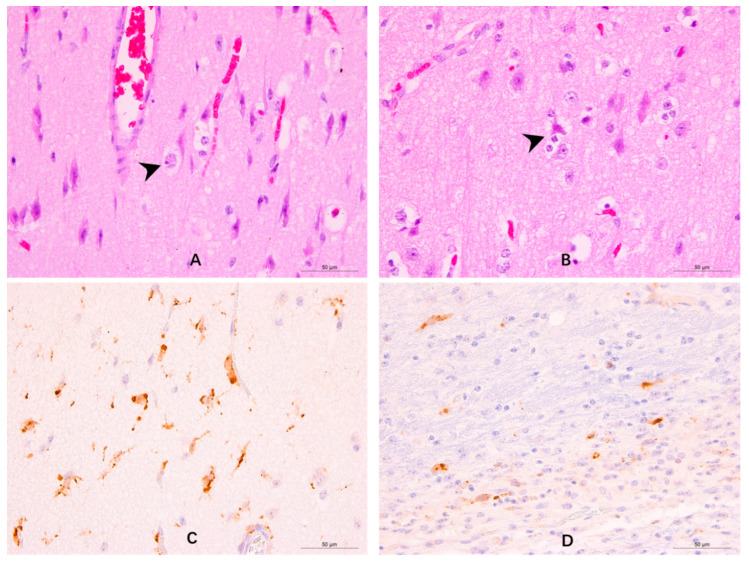
Brain tissue sections of a raccoon dog that died from CDV infection while displaying neurological symptoms [23]. Hematoxylin–eosin staining (**A**,**B**) and immunohistochemical analysis (**C**,**D**) of the tissues. Black arrows indicate cytonuclear eosinophilic inclusion bodies in neurons (**A**) and neuronophagia in the cerebral cortex (**B**). CDV-NP proteins (brown staining) were detected in neuronal and glial cells (**C**) and olfactory ensheathing cells (**D**).

**Table 1 viruses-14-01520-t001:** Comparison of symptoms and pathological changes in acute and late stages of CDV infection.

Infection Phase	Clinical Manifestations	Neurological Symptoms	Pathology of Non-Neuronal Tissues	Pathology of Nervous Distemper
Acute phase	Cutaneous rash; Serous nasal and ocular discharge; Conjunctivitis; Anorexia	Myoclonus; Nystagmus; Ataxia; Tetraparesis or plegia	Cytoplasmic and intranuclear inclusion bodies; Mucopurulent rhinitis; Interstitial pneumonia; Necrotizing bronchiolitis; Catarrhal enteritis; Hyper- and parakeratosis	Neuronal necrosis; Intranuclear inclusion bodies in neurons and astrocytes; Focal vacuolization of the white matter; Mild gliosis
Late phase	Subtle early clinical signs	Persistent myoclonus; CNS disturbances	Suppurative bronchopneumonia	Progressive perivascular mononuclear infiltrations; Astrogliosis

## Data Availability

Not applicable.

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
