# Peer review of "Multiple Receptors Involved in Invasion and Neuropathogenicity of Canine Distemper Virus: A Review"

_viruses, 2022, doi:10.3390/v14071520_

Round 1
Reviewer 1 Report
In this review article, the authors discuss the neurological consequence of canine distemper virus (CDV), with an emphasis on the route of invasion and progression. The authors also raise questions about the potential receptor(s) involved.
One of the major weaknesses is the choice of references and the lack of discussion on the role of different receptors between feral and lab-grown CDV. Some referenced studies used Onderstepoort strains so generalization should not be taken lightly.
This being said, I do find the title misleading because it is not clear whether invasion and/or neuropathogenesis are receptor-dependent. In any case, the authors merely described some receptors. A more thorough discussion of the use of different receptors by different CDV genotypes is needed, including potential vaccines.
Specific comments:
I would strongly suggest the authors acknowledge primary publications when possible (some examples below). Similarly, some statements are not supported by the reference provided.
Other mechanisms of CNS invasion are not mentioned or briefly touched on. For instance, consider infection of nectin-4 positive astrocytes cells via another mechanism such as promiscuous fusion triggering (PMID 30728259) or trans-endocytosis. (PMID 31331966)
Discuss the use of alternative receptors for CDV genotypes, as well as differences in pathogenicity. Some of the receptors discussed were published using the Onderstepoort strain, which is known to use alternative receptors to feral isolates.
Line 22 Typo: particularly
Line 32: Incorrect past tense use.
Line 48-49: Please rephrase
Lines 56 to 63: Reference?
Line 57: change “entering” for “enters”.
Line 69: Typo: mediated
Line 86: Typo: infection?
Line 87: Separate word: Are apparent.
Line 110 to 113, reference?
Lines 116-117: the static nature of the H protein is controversial, at least.
Line 121: Be specific, where is the front?
Line 162: Separate words: nectin like
Line 171: Reference 12 does not seem appropriate, PMID 24725937 does.
Lines 177-181. Reference 21 does not claim Nectin-4 expression in astrocytes. Therefore the word “In contrast” is misleading.
Line 191: Reference for CDV F binding to heparin sulfate?
Lines 200 to 201. Be more specific
Lines 205 to 208. CD46 is a receptor only for laboratory-adapted and attenuated measles viruses. This should be made clear.
Line 207. Substitute by primary references.
Line 208. CD46 is not a receptor for feral or lab-adapted CDV (PMID 25171206, 33534834)
Line 218-219. Rephrase, Vero cells don’t express nectin-4.
Line 225-238. Discuss the role of hyper fusogenic F proteins, as well as why not all infections lead to invasion of CNS. Any strain-specific differences?
Line 281. Reference for infection of neurons in measles SSPE?
Line 302. Oligodendrocytes can be infected.
Line 349. Reference 63 does not discuss MeV.
Author Response
Response to Reviewer 1 Comments
Point 1: In this review article, the authors discuss the neurological consequence of canine distemper virus (CDV), with an emphasis on the route of invasion and progression. The authors also raise questions about the potential receptor(s) involved.
Response 1: Yes. In this review, we mainly summarize the neuroreceptors and neuropathogenesis of CDV.
Point 2: One of the major weaknesses is the choice of references and the lack of discussion on the role of different receptors between feral and lab-grown CDV. Some referenced studies used Onderstepoort strains so generalization should not be taken lightly.
This being said, I do find the title misleading because it is not clear whether invasion and/or neuropathogenesis are receptor-dependent. In any case, the authors merely described some receptors. A more thorough discussion of the use of different receptors by different CDV genotypes is needed, including potential vaccines.
Response 2: Thanks to your suggestion, we have revised the erroneously cited references.
In addition, we revised the discussion on the role of receptors between wild-type and lab-grown CDV and reviewed the variations of invasion and neuropathogenicity between the different genotypes of CDV. (Page1: line2-3; Page7:268-270; Page7: line298-317; Page9: line366-415; Page5: line184; Page6: line220; Page11: line443 )
Specific comments:
Point 3: I would strongly suggest the authors acknowledge primary publications when possible (some examples below). Similarly, some statements are not supported by the reference provided.
Response 3: Thank you very much for your suggestion, we have corrected the incorrect reference citation in the article. (Page5: line184;Page6: line220;Page11: line443)
Point 4: Other mechanisms of CNS invasion are not mentioned or briefly touched on. For instance, consider infection of nectin-4 positive astrocytes cells via another mechanism such as promiscuous fusion triggering (PMID 30728259) or trans-endocytosis. (PMID 31331966)
Response 4: Thank you very much for the references, we have added additional mechanisms of neuronal infection. (Page6: line252-270)
Point 5: Discuss the use of alternative receptors for CDV genotypes, as well as differences in pathogenicity. Some of the receptors discussed were published using the Onderstepoort strain, which is known to use alternative receptors to feral isolates.
Response 5: Thanks to your suggestion, we have incorporated into the manuscript a comparison of the pathogenicity of CDV wild strains as well as vaccine strains. (Page7: line298-317; Page9: line366-415)
Point 6: Line 22 Typo: particularly
Response 6: Thanks for the reminder, we fixed this typo. (Page1: line22)
Point 7: Line 32: Incorrect past tense use.
Response 7: Thanks for your reminder, we have revised the tense of this sentence.(Page1: line32)
Point 8: Line 48-49: Please rephrase
Response 8: Thanks for the reminder, we've edited the sentence so it's more coherent.(Page2: line47-49)
Point 9: Lines 56 to 63: Reference?
Response 9: Thanks for your suggestion, we have completed the references. (Page2: line56-59)
Point 10: Line 57: change “entering” for “enters”.
Response 10: Thank you very much for the reminder that we have revised the tense of the word.(Page2: line57)
Point 11: Line 69: Typo: mediated
Response 11: Thanks for the reminder, we fixed this typo.(Page2: line69)
Point 12: Line 86: Typo: infection?
Response 12: Thanks for the reminder, we fixed this typo.(Page3: line86)
Point 13: Line 87: Separate word: Are apparent.
Response 13: Thanks for the reminder, we fixed this typo. (Page3: line87)
Point 14: Line 110 to 113, reference?
Response 15: Thanks for your suggestion, we have completed the references.(Page4: line121-123)
Point 15: Lines 116-117: the static nature of the H protein is controversial, at least.
Response 15: Thanks for your reminder, we have removed this ambiguous sentence.(Page4: line125-127)
Point 16: Line 121: Be specific, where is the front?
Response 16: Thanks for your suggestion, we have completed this section.(Page4: line129-134)
Point 17: Line 162: Separate words: nectin like
Response 17: Thanks for the reminder, we fixed this typo.(Page5: line174)
Point 18: Line 171: Reference 12 does not seem appropriate, PMID 24725937 does.
Response 18: I totally agree with your suggestion, this reference is more appropriate.(Page5: line184)
Point 19: Lines 177-181. Reference 21 does not claim Nectin-4 expression in astrocytes. Therefore the word “In contrast” is misleading.
Response 19: Thank you very much for the reminder, we have revised the sentence to be more coherent and less misleading. (Page5: line189-192)
Point 20: Line 191: Reference for CDV F binding to heparin sulfate?
Response 20: Thanks for the reminder, we have supplemented the references in this section.(Page5: line203)
Point 21: Lines 200 to 201. Be more specific
Response 21: Thank you very much for your suggestion, we have explained the specific role of CD9.(Page5: line208-214)
Point 22: Lines 205 to 208. CD46 is a receptor only for laboratory-adapted and attenuated measles viruses. This should be made clear.
Response 22: Thanks a lot for your suggestion, we added this detail to avoid ambiguity.(Page6: line218-222)
Point 23: Line 207. Substitute by primary references.
Response 23: Thanks for your suggestion, we have revised the references here.(Page6: line220)
Point 24: Line 208. CD46 is not a receptor for feral or lab-adapted CDV (PMID 25171206, 33534834)
Response 24: Thank you very much for your reminder that we have read the literature you provided and made changes to this section.(Page6: line220-222)
Point 25: Line 218-219. Rephrase, Vero cells don’t express nectin-4.
Response 25: Thanks for your suggestion, we have revised the sentence to be less misleading.(Page6: line230-233)
Point 26: Line 225-238. Discuss the role of hyper fusogenic F proteins, as well as why not all infections lead to invasion of CNS. Any strain-specific differences?
Response 26: Thanks for your suggestion, we have added relevant content to the manuscript. (Page6: line263-269; Page7: line298-317; Page9: line366-415; Page13: line558-576)
Point 27: Line 281. Reference for infection of neurons in measles SSPE?
Response 27: Thanks for your suggestion, we have added references on this part. (Page8: line335)
Point 28: Line 302. Oligodendrocytes can be infected.
Response 28: Yes, we strongly agree with you, we have revised the sentence to reduce the ambiguity. (Page9: line355)
Point 29: Line 349. Reference 63 does not discuss MeV.
Response 29: Thank you, we have revised the references here.(Page11: line443)

Reviewer 2 Report
Zhao and Ren provided a comprehensive review about receptor usage and neuropathogenesis of canine distemper virus (CDV). The manuscript is well written and contains all important aspects.
As many findings were compared later in the review with measles and its pathogenicity, it might be appropriate to say a few words about similarities and differences between measles and canine distemper in the introduction.
Line 181:The sentence "Although nectin-4 greatly contributes...." is misleading and can be deleted.
Figure 2: Epithelial cells are shown in step (1), but not macrophages and dentritic cells. Add these cells in the figure (between the epithelial cells). Furthermore, are the arrow from (3) to (4) and from (3) to (5) at the ight place? Doesn't the virus spread from step (2) to (4), and from outside to the olfactory nerve (5)?
Line 346: Instead of "take measels virus infection as an example" I would rather say: in the case of measles virus infection...
Line 371: The sentence may better read: Besides CDV and MeV, neurotropic viruses....including....have been identified.
Author Response
Response to Reviewer 2 Comments
Point 1: Zhao and Ren provided a comprehensive review about receptor usage and neuropathogenesis of canine distemper virus (CDV). The manuscript is well written and contains all important aspects.
Response 1: We appreciate the positive opinions of the reviewer. In this review, we mainly summarize the neuroreceptors and neuropathogenesis of CDV.
Point 2: As many findings were compared later in the review with measles and its pathogenicity, it might be appropriate to say a few words about similarities and differences between measles and canine distemper in the introduction.
Response 2: We very agree with your suggestion that we have added the relevant part of the measles virus to the introduction. (Page3: line104-112)
Point 3: Line 181:The sentence "Although nectin-4 greatly contributes...." is misleading and can be deleted.
Response 3: Thank you very much for your suggestion, we have removed this sentence and reduced misleading of this sentences. (Page5: line192)
Point 4: Figure 2: Epithelial cells are shown in step (1), but not macrophages and dentritic cells. Add these cells in the figure (between the epithelial cells). Furthermore, are the arrow from (3) to (4) and from (3) to (5) at the ight place? Doesn't the virus spread from step (2) to (4), and from outside to the olfactory nerve (5)?
Response 4: Thanks for your suggestion, we have edited the image. (Page8: line318)
Point 5: Line 346: Instead of "take measles virus infection as an example" I would rather say: in the case of measles virus infection...
Response 5: Thank you very much for your suggestion, the manuscript is more coherent after revision. (Page11: line440)
Point 6: Line 371: The sentence may better read: Besides CDV and MeV, neurotropic viruses....including....have been identified.
Response 6: Thank you very much for your suggestion, the manuscript is more coherent after revision. (Page11: line464-467)
Round 2
Reviewer 1 Report
I strongly recommend the authors to seek proof-editing services. Some of the new sections are incomprehensive/incoherent.
Line 113. The Edmonston strain has not been found spreading between neurons. SSPE cases have always been caused by wild-type MeV genotypes, and vaccine strains (genotype A) have never been encountered. (PMID 29466428).
Line 219-220. Rephrase and be concise. Viral receptors determine whether a cell is susceptible, but not whether it is permissive.
Lines 210-214. Be concise.
Lines 269-271. This statement is misleading. Remove. SSPE cases have always been caused by wild-type MeV genotypes,
Line 282. Substitute strain by genotype.
Line 274. Comma used instead of full stop.
Line 300. Genotype strains do not exist
Line 301 and 304. 5804P and A75-17 are not strains.
Line 304. What last? Neurological symptoms?
Line 305. Full stop before despite?
Line 309-311. I don’t understand the sentence.
Line 316-318. Rephrase.
Lines 371. Full stop before lesions.
Line 373. Incorrect use of linking word.
Line 388. Separate virus neutralizing.
Line 392: remove “the”.
Line 396: “a” variety.
Line 397: residues
Line 399; Remove “the”.
Line 400-402. Not sure what the authors tried to convey. D540G enable use of human SLAM by a specific CDV isolate.
Line 405. I don’t understand this sentence.
Line 407. Strain should be plural. In any case, it should say isolates.
Lines 409-416. Rephrase. Consider the following publications (PMID 17686846, 15893783, 11932382, 33534834).
Line 559-575. Revise grammar and meaning. Intracellular fusion is dependent on hemagglutinin and fusion proteins (PMID 29298883).
Author Response
Response to Reviewer Comments
Point 1: I strongly recommend the authors to seek proof-editing services. Some of the new sections are incomprehensive/incoherent.
Response 1: Thank you for your comment; the manuscript has now been proof-edited by a professional service (Editage company).
Point 2: Line 113. The Edmonston strain has not been found spreading between neurons. SSPE cases have always been caused by wild-type MeV genotypes, and vaccine strains (genotype A) have never been encountered. (PMID 29466428).
Response 2: Thank you for your comment and the reference; we have corrected the text accordingly (Page 3: lines 111–115).
Point 3: Line 219-220. Rephrase and be concise. Viral receptors determine whether a cell is susceptible, but not whether it is permissive.
Response 3: We very much agree that viral replication in a cell is not receptor-dependent, and we have revised the sentence accordingly (Page 5: lines 213–214).
Point 4: Lines 210-214. Be concise.
Response 4: Thank you for your suggestion; we have simplified this section accordingly (Page 5: lines 208–210).
Point 5: Lines 269-271. This statement is misleading. Remove. SSPE cases have always been caused by wild-type MeV genotypes.
Response 5: Thank you for your suggestion; we strongly agree with you and have deleted the indicated sentence (Page 6: lines 263–265).
Point 6: Line 282. Substitute strain by genotype.
Response 6: Thank you for your comment; we have revised as suggested (Page 7: line 277).
Point 7: Line 274. Comma used instead of full stop.
Response 7: Thank you for your comment; we have revised as suggested (Page 6: line 268).
Point 8: Line 300. Genotype strains do not exist
Response 8: Thank you for your comment; we have revised this sentence accordingly (Page 7: line 295).
Point 9: Line 301 and 304. 5804P and A75-17 are not strains.
Response 9: Thank you for your comment; we have revised this sentence to avoid ambiguity (Page 7: lines 296–300).
Point 10: Line 304. What last? Neurological symptoms?
Response 10: Thank you for your comment; we were making the point that the CDV infection persists for approximately 5 weeks. We have elaborated to clarify (Page 7: lines299).
Point 11: Line 305. Full stop before despite?
Response 11: Thank you for your comment; we have revised accordingly (Page 7: line 300).
Point 12: Line 309-311. I don’t understand the sentence.
Response 12: Thank you for your comment; we have reworked the sentence for clarity (Page 7: lines 301–305).
Point 13: Line 316-318. Rephrase.
Response 13: Thank you for your suggestion; we have revised this part of the sentence to improve the flow (Page 7: lines 310–313).
Point 14: Lines 371. Full stop before lesions.
Response 14: Thank you for your comment; we have revised accordingly (Page 10: line 366).
Point 15: Line 373. Incorrect use of linking word.
Response 15: Thank you for your comment; we have revised accordingly (Page 10: line 368).
Point 16: Line 388. Separate virus neutralizing.
Response 16: Thank you for your comment; we have revised as suggested (Page 10: line 384).
Point 17: Line 392: remove “the”.
Response 17: Thank you for your comment; we have revised as suggested (Page 10: line 389).
Point 18: Line 396: “a” variety.
Response 18: Thank you for your comment; we have revised accordingly (Page 10: line 396).
Point 19: Line 397: residues
Response 19: Thank you for your comment; we have revised as suggested (Page 10: line 393).
Point 20: Line 399; Remove “the”.
Response 20: Thank you for your comment; we have revised accordingly (Page 10: line 397).
Point 21: Line 400-402. Not sure what the authors tried to convey. D540G enable use of human SLAM by a specific CDV isolate.
Response 21: We wanted to say that the CDV H protein determines its virulence and tropism in different hosts, so that substitutions (D540G) in the H protein alter its fusion activity and viral adaptation to the host. We have now revised the text to explain this (Page 10: lines 398–400).
Point 22: Line 405. I don’t understand this sentence.
Response 22: We have now revised the sentence for clarity (Page 10: lines 400–403).
Point 23: Line 407. Strain should be plural. In any case, it should say isolates.
Response 23: Thank you for your suggestion; we very much agree, and we have revised the text accordingly (Page 10: line 404).
Point 24: Lines 409-416. Rephrase. Consider the following publications (PMID 17686846, 15893783, 11932382, 33534834).
Response 24: Thank you for your comment; we have revised the text based on the references you provided (Page 10: lines 405–408).
Point 25: Line 559-575. Revise grammar and meaning. Intracellular fusion is dependent on hemagglutinin and fusion proteins (PMID 29298883).
Response 25: Thank you for your suggestion; we have revised the text based on the references you provided and also for clarity (Page 13: lines 556–563).

This manuscript is a resubmission of an earlier submission. The following is a list of the peer review reports and author responses from that submission.
Round 1
Reviewer 1 Report
1) This paper overviews neurotropic viruses utilizing CDV neural receptors.
2) Some of the references are more than 10 years old, if authors can revise them to most current 5 years citations?
Tatsuo, H.; Ono, N.; Yanagi, Y., Morbilliviruses use signaling lymphocyte activation molecules (CD150) as 498 cellular receptors. Journal of virology 2001, 75, (13), 5842-50
3) Can authors split the sections in introduction as mentioned following 3 sections?
(1) the research progress on CDV receptors and their role in CD pathogenesis; (2) CDV invasion of the nervous system and the underlying pathogenic mechanism; and (3) possible neural receptors of CDV.
4) There are too many simple details in section 2. CDV entry hosts routes, these can be removed?
5) Can authors present a table comparing acute and late phase of CDV infection?
6) Authors need to provide citation for Figure 2. Brain tissue sections of a raccoon dog that died from CDV infection displaying neurological symptoms.
7) Its a well written article, there are some unnecessary details in introduction if authors could remove them and be concise on statements would benefit the readers. Thanks,
Reviewer 2 Report
In this article Zhao and Ren reviewed the mechanism of CDV infection, especially the mechanism of infection of the central nervous system. This review also summarizes the various human viruses that infect the central nervous system. This is a rather extensive review, which somewhat makes the subject matter difficult to understand in some respects. This review article may be structured and written in such a way that it may be difficult for readers who are not familiar with this topic, because it contains consensus issues and many other issues that may be still hypotheses of some researchers in the same line. It would become a much better review if the non-main agenda was stripped away and the focus was solely on the neuropathogenicity of CDV. Overall, I think a more plain style of writing would be better, and it would be a good review if it were structured in a more compact way. However, there have been no significant recent developments in the topics covered in this review. In fact, of the 94 papers cited in this review, only six are on CDV beyond 2019, and only one is beyond 2020. Probably due to these reasons, this review may have failed to generate any new perspectives or provide any particularly good commentary.
Specific comments
Major comments
- Throughout the text, many citations refer to "recent" research, but there are many studies that are more than 10 years ago. What kind of research is described as "recent" research?
- The "possible neural receptors of CDV" is one of the subjects of this review, but it is mostly described as unknown, with almost no new descriptions or discoveries of progress. Line 63
- Is it necessary to describe the detailed mechanism of membrane fusion by H and F in this review? In addition, the text is very difficult to understand, and there are some incorrect descriptions. Even if this chapter is retained, I think it needs to be revised substantially. Line 93-125
- There are several sections that described CDV-DL strain, but it is difficult to know whether the descriptions in the related sentances are descriptions of characteristic matters found in CDV-DL strain or matters commonly observed in CDV.
- Is the detailed description of MV, including the description of CD46, between lines 280 and 301 necessary?
- In the Future perspectives section, there are descriptions of issues and perspectives that have been repeatedly mentioned by many researchers over the years, but there are no clear descriptions of new and distinctive perspectives.
Minor comments
- Although incidents of CDV infection have indeed been confirmed in animals of Artiodactyla, Primates, Rodentia, Proboscidae etc., their susceptibility should be very different from that of carnivores. It may not be appropriate to say that the animals in these cases, which seem to be the result of spill over events, are also susceptible to CDV? Line 38-39
- The authors say "we speculate," but a great many researchers have been suggesting for quite some time that there are other receptors for CDV and other morbilliviruses in the central nervous system. Line 61
- Does signaling by H-protein binding also occur in the case of morbilliviruses? What kind of events are described here? It needs to be stated more clearly and also cited. Line 127-129
- There is a mention of the CEF, but it may be somewhat abrupt and its significance is difficult to understand in this article. Line 149-152
- The term "SLAM receptor" may be misunderstood as "a receptor for SLAM," so it is better to simply use "SLAM" or "receptor, SLAM,".
- Why does this paragraph dare to say "viral protein" instead of "H protein"? Line 157-177
- What exactly does "blocking interaction" mean? Line 167-168
- What kind of situation does the "strong in vivo selection" described here mean, and does such selection occur with natural infection with CDV? Line 171-174
- Is CD155 correct for nectin-5? Line 182
- Probably similar results are expected for MV and CDV, but the apical release papers cited are for MV, not for CDV. Line 192 (References 48, 49)
- Is it a recombinant CDV with some characteristic properties? Line 211
- Are there any clear cases of influenza viruses infecting the central nervous system? Line 369, 399
- Is there a consensus on the neurological infection of SAR-CoV-2? Line 412